# Emergency of Tsallis statistics in fractal networks

**Airton Deppman**[1]*, **Evandro Oliveira Andrade-II**[2]

**1** Instituto de Física, Universidade de São Paulo, São Paulo, SP, Brazil, **2** Departamento de Física, Universidade Estadual de Santa Cruz, Ilhéus, BA, Brazil

* deppman@usp.br

## Abstract

Scale-free networks constitute a fast-developing field that has already provided us with important tools to understand natural and social phenomena. From biological systems to environmental modifications, from quantum fields to high energy collisions, or from the number of contacts one person has, on average, to the flux of vehicles in the streets of urban centres, all these complex, non-linear problems are better understood under the light of the scale-free network's properties. A few mechanisms have been found to explain the emergence of scale invariance in complex networks, and here we discuss a mechanism based on the way information is locally spread among agents in a scale-free network. We show that the correct description of the information dynamics is given in terms of the q-exponential function, with the power-law behaviour arising in the asymptotic limit. This result shows that the best statistical approach to the information dynamics is given by Tsallis Statistics. We discuss the main properties of the information spreading process in the network and analyse the role and behaviour of some of the parameters as the number of agents increases. The different mechanisms for optimization of the information spread are discussed.

## 1 Introduction

A large number of problems that are common to modern societies can be addressed in the framework of complex networks. Accurate data and methods were made available by new technologies that are used worldwide, providing for the first time the adequate conditions to the development of scientific approaches to those problems. As a consequence, the last decades witnessed the fast evolution of our knowledge on the behaviour and properties of complex networks [1, 2].

In this work, we describe and prove some of the most important characteristics of the flux of information in a fractal network. Information, here, is considered in a broad sense, and can refer to pieces of information locally transmitted, or to people or objects that move from one node to another in the network. We say the information is locally transmitted because we consider only those cases where the information is transmitted from one agent to a limited and small number of agents in the same network. These characteristics are present in several of our socioeconomic activities, and in natural systems. We discuss that the variables describing the

**Data Availability Statement:** All relevant data are within the manuscript.

**Funding:** AD, grant 304244/2018-0, Conselho Nacional de Desenvolvimento Científico e Tecnológico, http://www.cnpq.br, did not play any

role in the design, analysis, prepration or submission of the work. AD, grant 2016/17612-7, Fundação de Amparo à Pesquisa do Estado de São Paulo, https://fapesp.br, did not play any role in the design, analysis, prepration or submission of the work.

**Competing interests:** The authors have declared that no competing interests exist.

different quantities in the system must appear in a scale-free form. In particular, we show that the time spent to share information among agents in the network is proportional to the squared-root of the number of agents. We argue that the scaling symmetry is broken at some point, and due to this symmetry break the information spreading follows a q-exponential function, and the statistical aspects of the network are hereby associated with the Tsallis Statistics [3].

Scale-free networks are of particular interest [4, 5] since many aspects of physical, biological and sociological systems [6–10] can be described to a good approximation by networks belonging to this class [11]. Several mechanisms for the emergence of scaling symmetry in complex networks have been identified [12] and comprehensive reviews on the subject can be found in Refs [12, 13]. The self-similarity in complex networks are associated to the observation of the power-law behaviour of the investigated quantity distribution. Such distributions are found in many places, as in biological systems [6, 14]; Internet structure [15].

In recent years, the ubiquity of scale-free networks in natural and social environments have been questioned [16, 17], even in places where it was considered to be frequently found [14]. This happens because other heavy-tailed distributions, different from the simple power-law, can also fit the data available. It is important to notice that, when referring to self-similarity in the realms of complex network, one basically mean those that follow a power-law distribution [18].

Scaling symmetry and a fine complex structure are the main features of fractals, so a fractal network also presents scale-free distributions and self-similarity [19], with the fractal dimension being related to topological characteristics of the system [20]. The constraints for fractal networks are stronger than for scale-free networks, with self-similarity, in the former systems, involving all relevant aspects of the network [18]. Despite being strongly constrained, fractal systems can be found in many places, as in high energy collisions in thermodynamics systems [21], Environmental Science [22, 23] or in Yang-Mills fields theory [24, 25], in the branching pattern of the circulatory system, in the metabolic rate of mammals [20]; in Epidemics [26–29]; in socioeconomic aspects of urban life, as in the distribution of wages or traffic of vehicles [30–32]. In this work, we address the fractal networks and the flux of information spreading on such class of networks. In particular, we show that the correct distribution to describe the information dynamics is the q-exponential behavior of the number of informed agents in the network, while the power-law behaviour appears in the asymptotic limit.

Important advances in the statistical analysis of the scale-free networks have been made [12, 33]. On the other hand, the statistical aspects of fractal systems are associated with the Tsallis Statistics [3], as it was shown in Ref. [21]. The q-exponential is a heavy-tailed function typical of the Tsallis statistics. The relation between information networks and Tsallis statistics has been a field of increasing interest in recent years [34, 35]. It is interesting to note that a fractal thermodynamics system, called thermofractal, presents some mathematical properties that are very similar to those that will be found in the present study. Recently, it was shown that the algebra of the group of transformation of thermofractals and the q-algebra [36] associated to Tsallis statistics, are isomorphic [37]. That conclusion can be applied to fractal networks as those studied in the present work. Remarkably, renormalization seems to be an essential feature in thermofractals [21, 38], Yang-Mills fields [25] (see [39] for a review on the subject) and in fractal networks [19]. The possibility of finding a common theoretical framework joining fractal networks and Tsallis statistics is an interesting subject of research.

The work is organized as follows: in Section 1 we describe the structure of the scale-free network, the scaling parameter and the break of the scaling symmetry; in Section 2 we describe the information spreading dynamics over the scale-free network, obtain the q-exponential behaviour for the spreading and show that any variable must follow a power-law distribution.

We obtain the differential equations that describe the information spreading dynamics and duscuss the different forms to increase the spreading efficiency. In Section 3 we discuss the main results and the possibilities to apply this theory to different problems and in Section 4 we present our conclusions.

## 2 Properties of the scale-free network

Fractal networks are represented by sets of nodes, connected among themselves. Each node in the network is itself a fractal network, similar to the initial one when its parameters are appropriately scaled [18]. This hierarchical structure is a prominent characteristic of fractal networks, organizing the agents in an undetermined number of levels. It is always possible to find a scaling parameter which allows describing any node at any level of the fractal structure by the same mathematical expressions and in terms of scale-free variables.

The structure of the fractal network used in the present work is depicted in Fig 1, where one individual, in red, has a piece of information that can be tranferred to other individuals in the same group, or agent. The dynamics of the information spreading in a small group can be scaled-up to larger groups using the renormalization properties of the fractal network [19].

These constraints on the network structure can be relaxed by the inclusion of probability distributions to determine some of the network features, but the distributions must be scale-free to preserve the scaling invariance. We will argue, in Section 4, that the results obtained here are very general and apply even in the case the constraints used here are relaxed.

In the following, we give definitions and derive some properties of a fractal network.

**Definition 1.1** *The fractal network is a set of N nodes, also called agents, totally or partially connected to the other nodes in the set. This network of N agents will be called main network.*

**Definition 1.2** *A scale-free, or fractal, network is a totally connected network. Each node is a fractal network similar to the main network, differing only by a scale parameter, forming an hierarchical structure. In this particular network, each agent is always comprised by N agents in the next level. For simplicity we will refer to the fractal network also by fnet.*

**Definition 1.3** *If an agent B is a component of an agent A, we say B is an internal agent of A, and that A is the parent agent of B.*

**Axiom 1.1** *The agents of a fractal network are connected exclusively to the agents in the same network and to their parent agents.*

**Axiom 1.2** *All the properties of the agents are constrained to keep the similarity of the fractal network.*

**Theorem 1.1** *The fractal network has a natural scale, λ, associated with the total number of agents in the fractal network.*

**Lemma 1.1.1** *The fractal network presents a hierarchical structure.*

*Proof:* From Definition 1.2 we observe that the *fnet* is composed of $N$ agents interconnected, and each one is itself a *fnet*, therefore each agent has its own internal structure. Due to Axiom 1.1, the internal structure of each internal agent is not connected to any other agent outside its network. This property establishes a level structure in the fractal network ranked according to the number of internal structures one needs to consider until a specific node is reached.

**Definition 1.4** *We say an agent is at a level l of the fractal structure if one needs to look into the internal structure of l agents, starting from one of the agents in the main fractal network, in order to find that agent. Agents at the mainfnetare at the level l = 1.*

**Corollary 1.1.1** *The total number of agents at the level l is $N^l$.*

*Proof:* Due to Definition 1.1 the*fnet*has $N$ agents at its first level. Due to Definition 1.2 each of those agents have $N$ internal agents, so at the second level, one has $N^2$. At each new level the number of agents is multiplied by $N$, so at the level $l$ the number of agents is $N^l$.

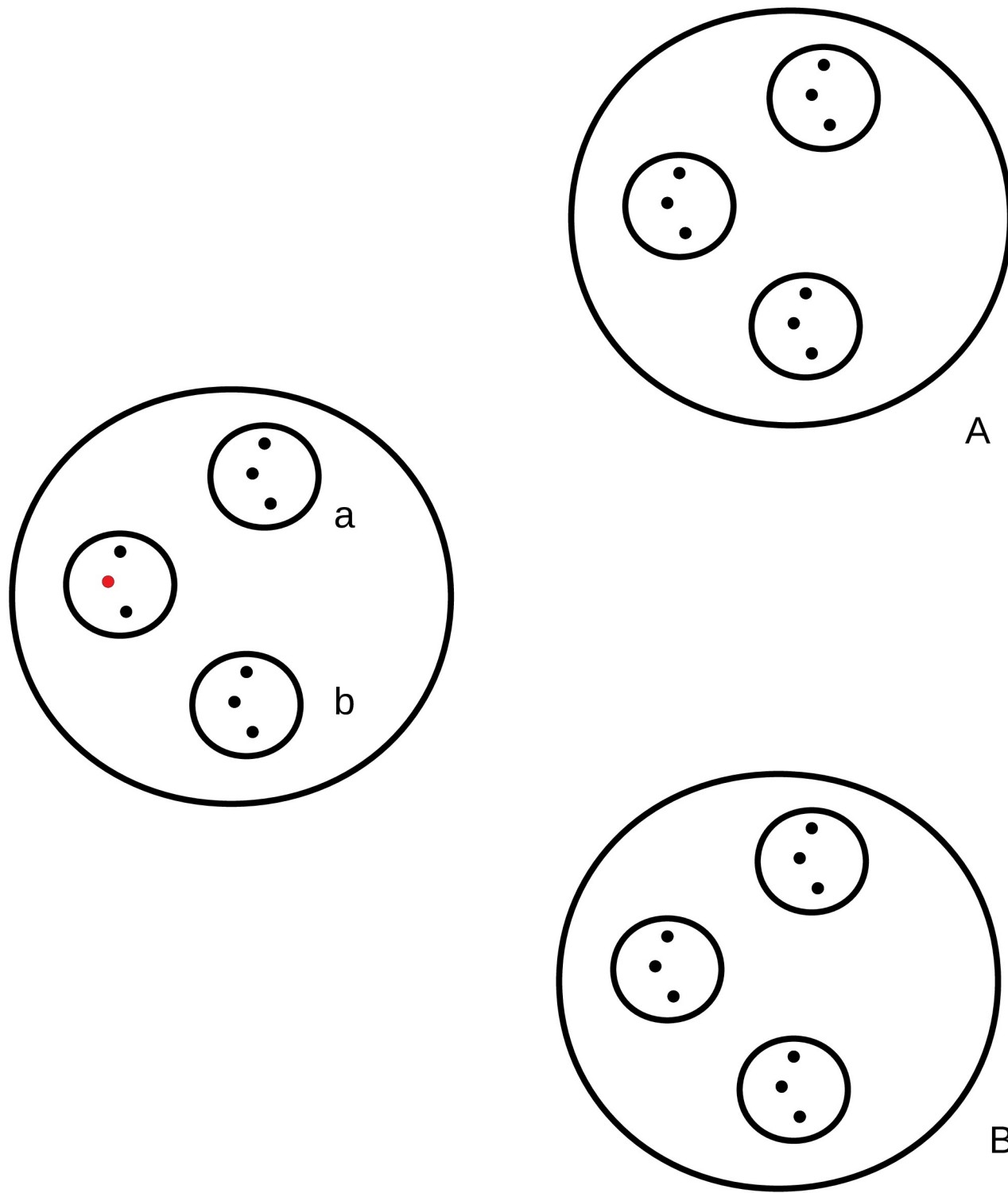

**Fig 1. Schematic view of the fractal network.** The individual represented by a red circle seeds the spread of information in the close contact group, or agent, represented by the smallest circle in which the individual is included. The spreading of information among larger groups, as indicated by letters a and b, scales according to the renormalization property of the fractal network. The same applies to still larger groups, as A and B.

**Definition 1.5** *The quantities that characterize the agents are of two types: or they are constant and are called parameters, or they are variables.*

**Corollary 1.1.2** *Any variable, v of the fnet must appear in the form $v/N^l$ and this ratio have to be scale invariant.*

*Proof:* Due to Axiom 1.2, the variables must scale according to some quantity that characterizes the agent size. The number of internal agents is a natural quantity to be used as a scaling parameter.

**Definition 1.6** *The size of the agents of a fnet can be unequivocally set by defining a level L at which the internal structure of the agents do not exist or can be disregarded in all relevant aspects of the fnet. The agents at this level are called individuals, and we refer to this level as the individual level.*

**Definition 1.7** *The scaling parameter, $\lambda_l$, that unequivocally determines the size of all agents in a fnet can be defined as $\lambda_l = N^{L-l}$, where L is the level of the individuals.*

**Lemma 1.1.2** *At the individual level the scaling symmetry is broken.*

*Proof:* This result follows immediately from Definition 1.6, since the individual agents do not have an internal structure.

**Corollary 1.1.3** *The number of internal individuals in an agent at the level l − 1 is given by $\sigma_{\lambda_l} = N\lambda_l$.*

*Proof:* It follows immediately from Definition 1.7.

**Corollary 1.1.4** *Since the scaling symmetry is valid at any level, aside from the individual level, the whole fnet can be seen as a single agent. Attributing the level l = 0 to the fnet allows one to represent the whole fnet as a single agent with $N^L$ internal individuals.*

*Proof:* It follows from Definition 1.7 and from Corollary 1.1.3.

This last result shows that there is a natural scale parameter, $\lambda$, that can be used to characterize the *fnet*, proving the Theorem 1.1.

## 3 Flux of information in a fractal network

In this section, we define what is meant by information and how it flows in the fractal network. The information is spread in the network from one initial agent that possess that piece of information, which is called an informed agent and transmits it or to the agents that are connected to it, or to its internal agents. When the piece of information reaches an uninformed agent it has a probability $\tau$ to received by that agent, and when it happens the uninformed agent becomes an informed one.

Information can be transmitted only by informed agents, and only uninformed agents can receive it. When an agent receives the piece of information, its first action is to transmit that piece of information to one of its internal agents. In the case of individual agents, it has a probability $\tau$ to change its state from uninformed to informed. When a fraction $\varphi$ of the internal agents are informed, the parent agent is considered informed.

Below we provide a formal description of the form of information spread in a fractal network and prove some of its characteristics. We also discuss some essential aspects of the scaling symmetry break, which will give rise to the q-exponential function and ultimately to the power-law behaviour.

**Definition 2.1** *Information is considered here in a broad sense, what includes pieces of information that can be shared among agents but also objects and people moving from one node to another.*

**Definition 2.2** *When an agent obtains the information, we say his states changes from uninformed to informed. The states of an agent are just informed or uninformed.*

**Axiom 2.1** *Agents can obtain the information only from informed agents with which they are connected, or from their parent agent. We say that, in this case, information is locally transmitted.*

**Axiom 2.2** *When an informed agent has a connection with an uninformed agent, we say that the information has reached the uninformed agent.*

**Axiom 2.3** *When a piece of information reaches an uninformed agent, it is passed to its internal agents aleatory chosen.*

**Axiom 2.4** *When a piece of information reaches an individual agent, it has a probability $\tau$ to accept that information. Upon acceptance, the individual becomes informed. The elapse of time to inform one individual is $\Delta t_o$.*

**Axiom 2.5** *An agent that is not an individual becomes informed when a fraction $\tau$ of its internal agents are informed.*

**Axiom 2.6** *Uninformed agents cannot transmit information. Informed agents necessarily transmit information.*

**Theorem 2.1** *If $\lambda_L$ is the scale at the individual level and $\sigma$ is the total number of individuals in the fnet, the number of informed agents, $v$, after the start of information transmission is given in terms of a q-exponential function of $\sigma$ as*

$$v(\sigma) = e_q(\tau\sigma)\,, \tag{1}$$

*where*

$$e_q(\tau\sigma) = \left[1 + (1-q)\frac{\tau\sigma}{\lambda_L}\right]^{\frac{1}{1-q}}, \tag{2}$$

*with*

$$1 - q = 1/N\,. \tag{3}$$

The parameter $q$ is called q-index, and is completely determined by $N$.

**Lemma 2.1.1** *The number of informed agents after a period of information transmission at a level of the fractal network is given by*

$$v(\tau) = (1 + \tau)^\alpha\,, \tag{4}$$

*Proof:* According to Axiom 2.1 and to Axiom 2.4 the information is exchanged by $N$ agents where, initially, one of the agents is informed and all others are uninformed. An uninformed agent can get the information by different modes: it can get it directly from the first informed agent in the network, or it can get it from other agents that got the information from the initial agent, or even by more indirect ways. Mathematically, the number of informed agents after a period of information transmission is given by

$$v(\tau) = 1 + \alpha\tau + (1/2)\,\alpha(\alpha-1)\tau^2 + \cdots + C(\alpha, k)\tau^k + \cdots + \tau^\alpha\,, \tag{5}$$

where $\alpha$ is the number of modes for the transmission of the information in the network. As one agent cannot transfer information to itself, we have that $\alpha = N - 1$.

The combinatorial factor

$$C(\alpha, k) = \frac{\alpha!}{(\alpha - k)!k!} \tag{6}$$

arises because when the information is transferred to an uninformed agent, the order in which

the informed agent obtained the information is not important. Eq 5 can be written as the power-law in Eq 4, proving this Lemma.

**Definition 2.3** *We denominate* $\sigma_{l,\lambda_{l'}}$, *with* $l' > l$, *the number of internal agents with size, or scale,* $\lambda_{l'}$ *that an agent at the level l has.*

**Corollary 2.1.1** *The number of informed agents can be expressed in terms of the scale parameter,* $\lambda_l$.

*Proof:* According to Corollary 1.1.3, each agent in a level $l-1$ has exactly $\sigma_{l-1,\lambda_l} = N\lambda_l$ internal agents with size $\lambda_l$. Therefore we can write $\tau = \tau\sigma_{l-1,\lambda_l}/(N\lambda_l)$. It follows immediately from Eq 4 that

$$v(\tau) = \left(1 + \tau \frac{\sigma_{l-1,\lambda_l}}{N\lambda_l}\right)^{N-1}. \tag{7}$$

**Corollary 2.1.2** *The ratio* $\sigma_{l,\,\lambda_{l'}}/\lambda_{l'}$ *is scale invariant.*

*Proof:* If you multiply $\lambda_{l'}$ by any positive, finite factor, due to Corollary 1.1.3 the number of agents at any level $l < l'$ is multiplied by the same factor, hence the ratio above remains invariant.

**Corollary 2.1.3** *The symbols* $\sigma_L$ *and* $\lambda_{L+1}$ *are meaningless.*

*Proof:* According to Definition 1.7, the level $L$ corresponds to that where agents are individuals, so they do not present an internal structure, hence there is no meaning in asking about its internal population. The individual size is the minimum size and determines the fundamental scale of the *fnet*, thereby there is no meaning in asking about scales below $\lambda_L$.

**Definition 2.4** *When the scale is set to the individual size, that is,* $\lambda_{l'} = \lambda_L$, *we use the simplified notation* $\sigma_l = \sigma_{l,\lambda_L}$. *Accordingly, we denote by* $\sigma$ *the population of individuals in the fnet, that is,* $\sigma = \sigma_0$.

**Corollary 2.1.4** *At the individual level we have* $\lambda_L = 1$, *the scale symmetry is broken and the q-exponential function is obtained.*

*Proof:* Adopting $\lambda_l = \lambda_L$, Eq 7 becomes

$$v(\tau) = \left(1 + \tau \frac{\sigma}{N\lambda_L}\right)^{N-1}, \tag{8}$$

where we used, for the sake of clarity, $\sigma = \sigma_{0,\lambda_L}$ as the total population of the network, recognized as a multiple the number of individuals in the network. Using the q-index defined by Eq (3) results that

$$v(\sigma) = \left[1 + (1-q)\frac{\tau\sigma}{\lambda_L}\right]^{\frac{q}{1-q}}. \tag{9}$$

The expression above is not scale invariant, because now $\lambda_l = \lambda_L$ is fixed, and any variation of the *fnet* population, $\sigma$, results in a q-exponential behaviour.

This result proves the Theorem 2.1. Notice that with the introduction of the individual level, indicated by the scale $\lambda_L$, the scale invariance disappears and we obtain according to the q-exponential function.

An additional comment is necessary at this point. Observe that the argument of the functions in Eqs 8 and 9 are different. This results from the transition from a scale-free network to a fixed scale network. In the first case, the population increases according to the size of $\lambda_{l'}$ and the Corollary 1.1.3 is satisfied. In the second case, due to the symmetry break, the population increases while the scale is fixed. The number of close contacts agent is also fixed, as well as the

parameter $q$. This means that the number of degrees of freedom for the information spread is independent of the population size. This is the main aspect of network for the emergence of non-extensivity, as will be discussed in Section 4.

## 3.1 The dynamics of the information spread

In this section, we describe how to describe the dynamics of the information spread by including the time evolution of the number of informed agents.

**Definition 2.5** *We define the information spread time interval, $\Delta t_\lambda$, and the rate of transmission of information, $\kappa_\lambda$, such that $\tau = \kappa_\lambda \Delta t_\lambda$.*

**Theorem 2.2** *The time interval $\Delta t_\lambda$ depends on the population size, $\sigma_\lambda$ as a power-law function, that is, $\Delta t_\lambda = \sigma^\beta \Delta t_{\lambda_L}$, and the transmission rate depends on the population size according to $\kappa_\lambda = \sigma_\lambda^{-\beta} \kappa_{\lambda_L}$.*

**Lemma 2.2.1** *The time interval of an agent, $\Delta t_{\lambda_l}$, is related to the internal agents time interval, $\Delta t_{\lambda_{l+1}}$ by $\Delta t_{\lambda_l} = N^\beta \Delta t_{\lambda_{l+1}}$, with $0 \leq \beta \leq 1$. The rate of information transmission are related by $\kappa_{\lambda_l} = N^{-\beta} \kappa_{\lambda_{l+1}}$.*

*Proof:* Consider an agent at the scale $\lambda_l \neq \lambda_L$. According to Definition 1.1 this agent has $N$ internal agents. The elapsed time for the information transmission to an agent, $\Delta t_{\lambda_l}$, depends on the time its internal agents will demand to get the information, $\Delta t_{\lambda_{l+1}}$.

Due to Axiom 1.2 all agents at the same hierarchic level corresponding to $\lambda_{l+1}$ have similar values for $\Delta t_{\lambda_{l+1}}$, so the maximum value for interval for the parent agent is $\Delta t_{\lambda_l} = N \Delta t_{\lambda_{l+1}}$ in the case of the information is spread among the internal agents sequentially. The minimum value is $\Delta t_{\lambda_l} = \Delta t_{\lambda_{l+1}}$, in the case of simultaneous transmission of the information among the internal agents. In the gernal case we write $\Delta t_{\lambda_l} = N^\beta \Delta t_{\lambda_{l+1}}$, with $0 \leq \beta \leq 1$. The equalities correspond to the two special cases mentioned above. As $\tau$ is a parameter, according to Definition 1.6 we must have $\kappa_\lambda = N^{-\beta} \kappa_{\lambda'}$.

**Corollary 2.2.1** *The parameter $\beta$ is independent of the agent level.*

*Proof:* It follows from the Axiom 1.2.

**Corollary 2.2.2** *The variables $\Delta t_{\lambda_1}$ of an agent at level $l_1$ is related to the variable $\Delta t_{\lambda_2}$ at the level $l_2 > l_1$ by $\Delta t_{\lambda_1} = N^{\beta(l_2 - l_1)} \Delta t_{\lambda_2}$, and the variable $\kappa_{\lambda_1}$ is related to $\kappa_{\lambda_2}$ by $\kappa_{\lambda_1} = N^{-\beta(l_2 - l_1)} \kappa_{\lambda_2}$.*

*Proof:* Applying recursively the result of Lemma 2.2.1 we get

$$\Delta t_{\lambda_1} = \left( \prod_{i=l_1}^{l_2} N_i^\beta \right) \Delta t_{\lambda_2} = N^{\beta(l_2 - l_1)} \Delta t_{\lambda_2}. \tag{10}$$

The result for $\kappa_{\lambda_1}$ can be obtained in the same way or by considering that $\tau$ is constant, as done in Section 3.1.

Considering the result of Corollary 2.2.2 for an agent at level $l_1 = l$ and another agent at level $l_2 = L$, we get

$$\begin{cases} \Delta t_{\lambda_l} = N^{\beta(L-l)} \Delta t_{\lambda_L} \\ \\ \kappa_{\lambda_l} = N^{-\beta(L-l)} \kappa_{\lambda_L} \end{cases}. \tag{11}$$

Setting $l = 0$ we have $N^L = \sigma$, that is the number of individuals in the *fnet*, these results prove Theorem 2.2.

**Theorem 2.3** For a random *fnet* $\beta = 0.5$.

*Proof:* The time interval, $\Delta t$, for the information spread for an agent with a number $\sigma$ of internal agents if $\delta t$, is formed by the superposition of the intervals $\delta t$ for the information transmission to each of the internal agents.

If the interval of transmission for $n - 1$ of the agents is $\Delta t_{n-1}$, the inclusion of an additional agent might increase the total time spent for information transmission only if the transmission in the $n$th agent starts at the instant $t$ such that $\Delta t - \delta t < t < \Delta t$. This condition is satisfied with a probability $\delta t / \Delta t_n$, and when it happens the increase in the total interval of time is $\delta t / 2$, on average, otherwise, the increase is null. Therefore we have, for $\sigma$ sufficiently large,

$$\sigma \frac{d\Delta t}{dn} = \frac{1}{2} \sigma \frac{\delta t}{\Delta t} \delta t \, . \tag{12}$$

If $\eta = n/\sigma$, we have $0 \leq \eta \leq 1$, and the equation above becomes

$$\frac{d\Delta t}{d\eta} = \frac{1}{2} \sigma \frac{\delta t}{\Delta t} \delta t \, . \tag{13}$$

Integrating from $\eta = 0$ to $\eta = 1$ we have

$$\Delta t = \sigma^{0.5} \delta t \, , \tag{14}$$

what proves that $\beta = 0.5$.

## 3.2 Differential equations for the information spread

In this section we derive the differential equations governing the dynamics of the information spreading. In what follows we assume that, at any time, the number of agents being informed is much smaller than the total population, therefore the variation of the uninformed population during the elapse of time necessary to the newly informed agents change their states from uninformed to informed is negligible. This can be expressed mathematically by assuming that $\dot{u} t \ll u$ at any time.

**Definition 2.6** *The number of uninformed individuals in a population of individuals, $u(t)$, varies along time as more individuals receive the information and become informed agents. The uninformed population at any time is given by $u(t) = \sigma - v(t)$, where $\sigma$ is the population in the fnet, which is considered constant.*

**Theorem 2.4** *Given a small time interval $\Delta t$, it is always possible to find an agent for which the elapsed time to spread the information is $dt < \Delta t$.*

*Proof:* Consider an arbitrary agent at a level $l_1$ whose spreading time is $\Delta t_{\lambda_1}$. If $\Delta t_{\lambda_1} < \delta t$, the condition is satisfied and the theorem is proved. If $\Delta t_{\lambda_1} > \delta t$, using Eq (10), one can find a level $l_2 > l_1$ at which the agents have a spreading time interval $\Delta t_{\lambda_2}$ such that

$$\Delta t_{\lambda_2} = N^{-\beta(l_2 - l_1)} \Delta t_{\lambda_1} < \delta t \, . \tag{15}$$

**Definition 2.7** *We call smooth information spreading dynamics the process for which the individual spreading time is sufficiently small, so that for any reasonably small time interval $\Delta t$, the elapsed time for an individual receive a piece of information, once the individual is reached by the spreading dynamics, is $\Delta t_{\lambda_L} < \delta t$.*

In what follows we assume the spreading dynamics is smooth.

**Theorem 2.5** *If at time t, measured in an appropriate scale for the dynamics of information at the individual level, afnethas $u(t)$ uninformed individuals, the rate of increase in the number*

*of informed individuals, i(t), is represented by* Eq 7, *which we write as*

$$\frac{di}{dt} = \kappa \frac{u(t)}{\lambda_L} \left[ 1 + (1-q) \frac{\tau u}{\lambda_L} \right]^{\frac{q}{1-q}}. \tag{16}$$

*Proof:* When a piece of information reaches an agent at the level $L - 1$, it is passed to its $N$ internal individuals. In a population, $u$, of uninformed individuals, the number of agents at this level is

$$M = \frac{u}{\lambda_L}, \tag{17}$$

because of Corollary 1.1.3. The number of those groups that receive the piece of information in the interval $dt$ is

$$dM = \kappa M dt. \tag{18}$$

For each group reached by the information, the number of individuals turning to the state informed is given by Eq 4, thus the number of individuals changing their state fron uninformed to informed is given by

$$di = \kappa M dt (1+\tau)^{N-1}. \tag{19}$$

Using Eq (3) and Definition 2.6 we obtain Eq (16), proving the theorem.

**Corollary 2.5.1** *The number of informed individuals in the network as a function of time is*

$$i(t) = \left[ 1 + (1-q) \frac{\kappa u(t) t}{\lambda_L} \right]^{\frac{1}{1-q}}. \tag{20}$$

*Proof:* Deriving Eq (20) and using the assumption $\ddot{u}t \ll u$, we obtain the differential equation given in Eq (16), proving the theorem.

**Corollary 2.5.2** *If a network is formed by N fnets with independent spreading dynamics, the number of informed individuals is*

$$i(t) = \sum_j i_j(t). \tag{21}$$

*where*

$$i_j(t) = \left[ 1 + (1-q) \frac{\kappa_j u_j(t - t_{oj}) t}{\lambda_L} \right]^{\frac{1}{1-q}}. \tag{22}$$

*if $t > t_{oj}$ and $i_j(t) = 0$ if $t < t_{oj}$. Here, $t_{oj}$ is the time when the information is received by the agent.*

*Proof:* It follows directly form Corollary 2.5.1.

**Theorem 2.6** *The spread of information in an agent can be described by the two coupled differential equations below (For the sake of clarity, we do not use the index j when we refer to the*

process in a single agent):

$$
\begin{cases}
\dfrac{di(t_{\lambda_L})}{dt_{\lambda_L}} = \dfrac{\kappa}{\lambda_L} i^q(t_{\lambda_L})\, u(t_{\lambda_L}) + \dfrac{\kappa}{\lambda_L} i^q(t_{\lambda_L})(t_{\lambda_L} - t_o)\dot{u}(t_{\lambda_L}) \\[2em]
\dfrac{du(t_{\lambda_L})}{dt_{\lambda_L}} = -\dfrac{\kappa}{\lambda_L} i^q(t_{\lambda_L})\, u(t_{\lambda_L})
\end{cases}
\tag{23}
$$

where $t_o$ is the instant when the spread of information starts.

*Proof:* By differentiating Eq (21) we obtain the first equation above. Considering that the uninformed population is determined according to Definition 2.6, the second equation is obtained.

**Theorem 2.7** *The set coupled differential equations can be written in terms of constant parameters in the case where the function u(t) can be linearized, that is, the coupled equation can be written as*

$$
\begin{cases}
\dfrac{di(t_{\lambda_L})}{dt_{\lambda_L}} = \dfrac{\kappa}{\lambda_L} i^q(t_{\lambda_L})\, u(t_{\lambda_L}) - \dfrac{\kappa^\dagger}{\lambda_L} i^q(t_{\lambda_L}) \\[2em]
\dfrac{du(t_{\lambda_L})}{dt_{\lambda_L}} = -\dfrac{\kappa}{\lambda_L} i^q(t_{\lambda_L})\, u(t_{\lambda_L})
\end{cases}
\tag{24}
$$

*where*

$$
\kappa^\dagger = \frac{\tau}{2}|\langle \dot{u} \rangle|,
\tag{25}
$$

*with $|\langle \dot{u} \rangle|$ being the modulus of the average rate of decrease of the uninformed population during the information spreading. is the instant when the spread of information starts.*

*Proof:* Considering the second term on the righ-hand side of the first equation, we have

$$
\kappa(t - t_o)\kappa i^q(t)u(t) = -\kappa(t - t_o)\dot{u}(t) \sim -\langle \kappa(t - t_o)\dot{u} \rangle.
\tag{26}
$$

We also have

$$
\langle \kappa(t - t_o) \rangle = \langle \kappa \Delta t \rangle \frac{\langle t - t_o \rangle}{\Delta t},
\tag{27}
$$

and we identify $\tau = \kappa \Delta t$. Considering that the distribution of new informed agents is practiacally symmetric with respect to the peak position, we also can approximate

$$
\frac{\langle t - t_o \rangle}{\Delta t} \sim \frac{1}{2}.
\tag{28}
$$

Using these results we prove the theorem.

**Theorem 2.8** *When the variation in the total population can be disregarded, the spread of information in a scale-free network can be approximately described by the two coupled*

*differential equations below*:

$$\begin{cases} \dfrac{di(t_{\lambda_L})}{dt_{\lambda_L}} = \dfrac{\kappa}{\lambda_L} i^q(t_{\lambda_L})\, u(t_{\lambda_L}) \\[20pt] \dfrac{du(t_{\lambda_L})}{dt_{\lambda_L}} = -\dfrac{\kappa}{\lambda_L} i^q(t_{\lambda_L})\, u(t_{\lambda_L}) \end{cases} \tag{29}$$

*where $t_o$ is the instant when the spread of information starts.*

*Proof:* If $|\langle \dot{u}\rangle| \sim 0$, then $\kappa^\dagger = 0$, and the result is evident from the last theorem.

**Theorem 2.9** *The solution to the coupled equations are*

$$\begin{cases} i(t_{\lambda_L}) = \left[1 + (1-q)\dfrac{\kappa u(t)(t_{\lambda_L} - t_o))}{\lambda_L}(t_{\lambda_L} - t_o)\right]^{1/(1-q)} \\[18pt] u(t_{\lambda_L}) = u(t_o) - i(t_{\lambda_L} - t_o) \end{cases} \tag{30}$$

**Theorem 2.10** *An approximate analytical solution for $u(t)$ can be obtained, resulting in*

$$u(t)(\theta_{\lambda_L}) = u(t_o)\left[1 + (1-q')\dfrac{\theta_{\lambda_L} - \theta_o}{\lambda'}\right]^{-1/(1-q')}, \tag{31}$$

*with*

$$\begin{cases} 1 - q' = \dfrac{q}{1-q} \\[14pt] \theta_{\lambda_L} = (\kappa t_{\lambda_L})^{2-q'} u_o^{1-q'} \\[14pt] \lambda' = (2-q')\lambda. \end{cases} \tag{32}$$

*Proof:*

An approximate solution can be easily obtained by noticing that, in most cases of interest, we have $u(t)/\lambda \gg 1$. In this case we can approximate the equation for $u(t)$ by

$$\dfrac{du(t)}{dt_{\lambda_L}} = -\dfrac{\kappa}{\lambda_L}\left[(1-q)\dfrac{\kappa(t_{\lambda_L} - t_o)}{\lambda_L}\right]^{q/(1-q)} u^{1/(1-q)} \tag{33}$$

that is a separable equation resulting in

$$\dfrac{du(t_{\lambda_L})}{u^{1/(1-q)}} = -\dfrac{1}{1-q}\left[(1-q)\dfrac{\kappa t_{\lambda_L}}{\lambda_L}\right]^{1/(1-q)} dt_{\lambda_L}. \tag{34}$$

Integrating both sides we get

$$\int_{u_o}^{u} \dfrac{du'}{u'^{1/(1-q)}} = -\dfrac{1}{1-q}\left[(1-q)\dfrac{\kappa}{\lambda_L}\right]^{1/(1-q)} \int_{t_o}^{t_{\lambda_L}} t'^{1/(1-q)} dt'. \tag{35}$$

This equation results in

$$u(t_{\lambda_L})^{-q/(1-q)} = u_o^{-q/(1-q)} + \dfrac{q}{1-q}\left[\left(\dfrac{(1-q)\kappa t_{\lambda_L}}{\lambda_L}\right)^{1/(1-q)} - \left(\dfrac{(1-q)\kappa t_o}{\lambda_L}\right)^{1/(1-q)}\right], \tag{36}$$

and can be rearranged to obtain

$$u(t_{\lambda_L}) = u_o \left\{ 1 + \frac{q}{(1-q)} \left[ \left( \frac{(1-q)\kappa u_o^q t_{\lambda_L}}{\lambda_L} \right)^{1/(1-q)} - \left( \frac{(1-q)\kappa u_o^q t_o}{\lambda_L} \right)^{1/(1-q)} \right] \right\}^{-\frac{1-q}{q}}. \tag{37}$$

Observe that with the definitions given in the Eq (32) the equation above can be conveniently written as Eq (31), proving the theorem.

## 3.3 Strategies for optimization of the information diffusion

One of the most important results of the investigation of flux of information through networks is the possibility to understand the optimization of the information spread dynamics, what is important both for increasing the efficiency of communication and for formulating the best methods to avoid the information spread.

The main characteristic of the dynamics of information spread in the fractal network studied here is the local transmission of information by a small number of agents with close contact. The question that arises is the following: what is the best way to increase the efficiency of the information spread?

Two mechanisms could be devised to increase the efficiency: improve the probability of transmission, described by the parameters $\tau$ or by $\kappa$, or increasing the number of contacts between agents, given by $q$. We will see that the second option, when available, is the most effective.

**Theorem 2.11** *The increase of the rate of information transmission by increasing the efficiency of transmission is given by*

$$i_q(\tau + \delta\tau) = \left[ 1 + \frac{u/\lambda_L}{1 + (1-q)\tau u/\lambda_L} \delta\tau \right] i_q(\tau) \tag{38}$$

*Proof:* Using Definition 2.5 in Eq (20) and deriving with respect to $\tau$ we have

$$\frac{di_q}{d\tau}(\tau) = i_q^q(\tau), \tag{39}$$

therefore the infinitesimal variation in the number of informed agents when the transmission probability varies from $\tau$ to $\tau + \Delta\tau$ is

$$\delta i_q(\tau) = i_q(\tau) \frac{u/\lambda_L}{1 + (1-q)\tau u/\lambda_L} \delta\tau \tag{40}$$

Hence, when the transformation $\tau \to \tau' = \tau + \delta\tau$ is performed, the number of informed agents transformation is

$$i_q(\tau) \to i_q(\tau + \delta tu) = \left[ 1 + \frac{u/\lambda_L}{1 + (1-q)\tau u/\lambda_L} \delta\tau \right] i_q(\tau). \tag{41}$$

**Corollary 2.11.1** *In the limit $\tau u/\lambda_L \gg 1/(1-q)$, the transformation of $\tau$ leads to a logarithmic increase in the number of informed agents.*

*Proof:* In this limit we have

$$\frac{u/\lambda_L}{1 + (1-q)\tau u/\lambda_L} \delta\tau \sim \frac{\delta\tau}{1 + (1-q)\tau} = \frac{1}{1-q} \delta\log\tau. \tag{42}$$

Substituting the result above in Eq (41) we obtain the logarithmic increase of the number of informed agents, i.e.,

$$i_q(\tau) \to i_q(\tau + \delta\tau) = \left(1 + \frac{1}{1-q}\delta\log\tau\right)i_q(\tau).$$ (43)

**Theorem 2.12** *The increase of the rate of information transmission by increasing the number of links per agent is*

$$i_{q-\delta q}(\tau) = \left(1 + \frac{q}{(1-q)^3}\delta q\right)i_{q-\delta q}(\tau)$$ (44)

*Proof:* Using Definition 2.5 in Eq (20), and deriving with respect to $N$ we have

$$\frac{di_N(\tau)}{dN} = (1+N)\delta N\, i_N(\tau).$$ (45)

From Eq (3) it follows that

$$\frac{di_q(\tau)}{dq} = -\frac{q}{(1-q)^3}i_q(\tau)$$ (46)

From the results above we obtain that, under the transformation $q \to q - \delta q$, the number of informed agents transforms as

$$i_q(\tau) \to i_{q-\delta q}(\tau) = \left[1 + \frac{q}{(1-q)^3}\delta q\right]i_q(\tau).$$ (47)

**Corollary 2.12.1** *The increase in the number of informed agents increases with $N^2$.*
*Proof:* Using Theorem 2.1 for the relation between $N$ and $q$, and the Theorem 2.12 we obtain

$$i_N(\tau) \to i_{N+\delta N}(\tau) = [1 + N\delta N]i_q(\tau).$$ (48)

## 4 Discussion of the results

The characteristics of the network presented in Section 1 lead to the formation of a hierarchical scale-free structure typical of scale-free, or fractal, networks. The scaling parameter, λ, is given in terms of the number of internal agents. At some points the agents are considered as individuals with no internal structure, and at this point the scaling symmetry is broken.

The locally transmitted information and its spreading dynamics is defined in Section 2. The information is always shared among a number of connected agents in the same network or with the parent agent. This number is limited and constant throughout the network. This characteristic of the information spread dynamics and the broken scaling symmetry of the network structure give rise to a q-exponential function that describes the flux of information in the network. The power-law behaviour is obtained asymptotically, as the number of agents in the network increases. The results obtained here contributes to the discussion about the ubiquity of scale-free networks, since we obtain a heavy-tailed distribution that is not, in general, a power-law.

The fact that the number of informed agents is described in terms of a q-exponential function, with the power-law behaviour obtained in the asymptotic limit, is an interesting result

and deserves some additional comments. The q-exponential function results from the fact that the number of degrees of freedom of the spreading dynamics [5] is uncorrelated to the number of agents in the network. This aspect of the fractal structure allows the number of individuals increase without any change in the number of modes by which an arbitrary agent can exchange information with another in the *fnet*. It is easy to understand, from the results obtained here, why *fnets* can describe so many aspects of natural and social systems: in many cases the information is transmitted locally among a small group of agents, and this number will be the same, no matter how many individuals one are in that population of the network.

The Theorems proved in Section 2 show that any variable describing some quantity related to the information spread must appear in a scale-free form. The time interval for the spread of information in the network, for instance, increases with the squared-root of the number of individuals in the network. This result is in agreement with the works in Refs. [12, 13], where the close connections between power-law distributions and scale-free networks is observed. In the present case, we show that any fractal network will depend only on power-law variables.

The rate of information spread given by the q-exponential function shows that the spread dynamics results in a slower transmission of information than one would expect in an exponential spread. But as $q \to 1$ the number of links among agents increases and the exponential behaviour is recovered. This corresponds to broadcast information, with chaotic transmission of information to all the individuals in the network. We verified that the most efficient way to increase the number of informed agents is not by increasing the transmission probability, but by increasing the number of connections among agents.

This result is interesting in many aspects, but here we would like to emphasize one of them with an example of application in an epidemic spread of a virus. The piece of information being transmitted is the virus, and the transmission happens in close contact between an infected individual and a susceptible individual, that is, one that does not carry that piece of information, the virus. Observe that the coupled equations in Theorem 2.7 are very similar to those found in the standard SI and SIR models [40], except that we did not consider in those equations the recovered population, what can be straightforward done by considering that the total population is the sum of each kid of population, infected, susceptible and recovered, and that this population is constant. In Fig 2 we show plots of the informed and uninformed population along time, as well as the number of individual that receives the information at each instant.

We can understand the aspects related to the best strategies for the spreading dynamics also in the context of contagious diseases. A virus undergoes random mutation, and the dominant strain will be more likely the one that can be transmitted more effectively. The way mutagenesis of virus can lead to a more effective spread is not by increasing its probability of transmission, which we associate with the parameter $\tau$, but by increasing the number of susceptible individuals in contact with the infected individuals, which we associate with $N$. Those strains that succeed to increase $N$ will be more effective in transmission, and therefore will be dominant. Thus, viruses will increase the multiplication factor more efficiently if they succeed to provide a longer transmission time before the symptoms of its associated disease become evident.

As mentioned in the introduction, the definitions given in Section 1 and in Section 2 can be relaxed in many ways. For instance, the number of internal agents can be set as variable, but must follow the same distribution whatever is the fractal level of the parent agent, and the corresponding variable must be scale-free, i.e., it must appear as fractions of the scaling parameter. The same reasoning applies to the number of edges linking the agents. The information spread can follow an arbitrary distribution instead of being completely random, as far as the distribution is scale-free. Even the number of modes, or degrees of freedom, by which the agents can

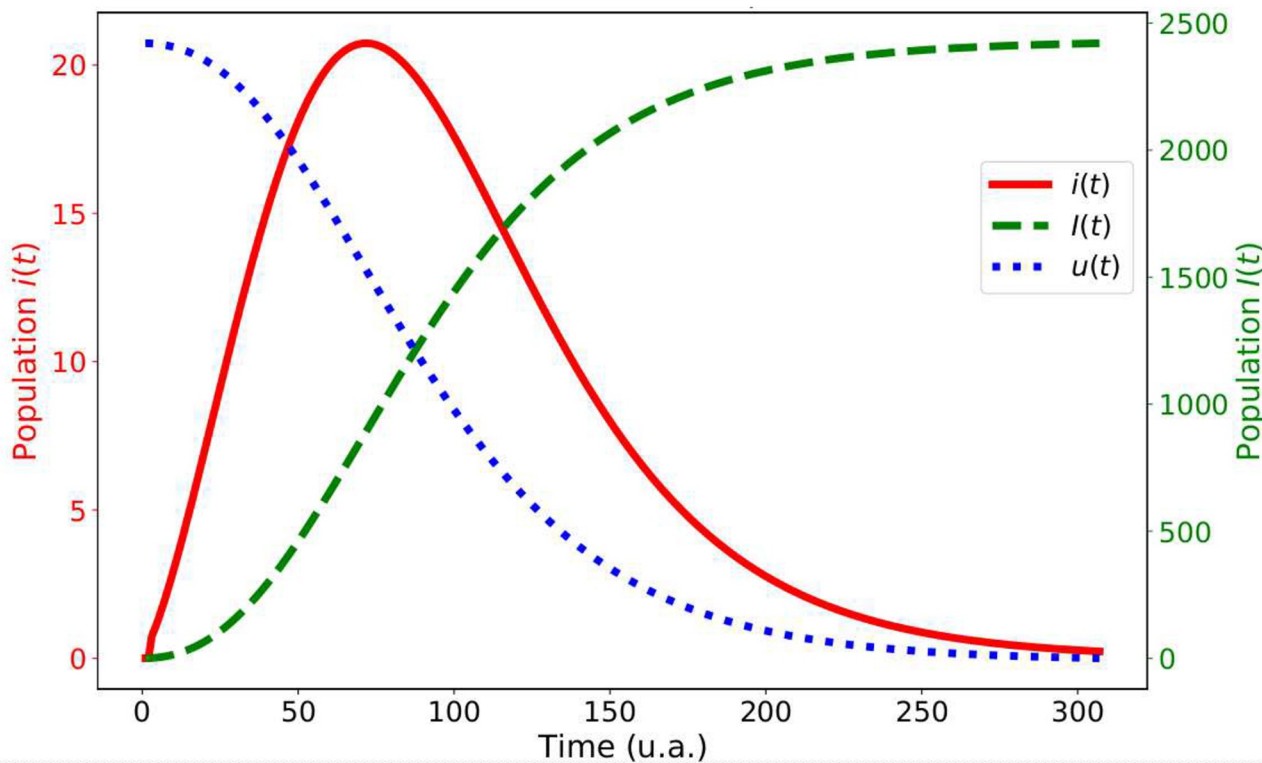

**Fig 2. Plots of the uninformed (dotted blue line), informed (dashed green line) and newly informed populations (continuous red line), according to the fractal dynamics for the information spread.**

obtain information from the others in the same network do not need to be constant. If these modifications are introduced in the scale-free network presented here, as far as the scale symmetry is preserved, our conclusions should hold. Even if different numbers of edges among the nodes are used throughout the network, a multifractal network may be obtained. In all these cases, however, the general results obtained here will remain valid.

## 5 Conclusion

In this work we studied the spreading dynamics of information locally transmitted through nodes, or agents, in self-similar, or fractal, network. The fractal network is defined by its fine internal structure that is scale-free. The self-similarity is a consequence of the the scaling property of the network and of its fine internal structure. However, at some point the scale symmetry is broken, and as a result the flux of information follows a q-exponential function, typical of the Tsallis' statistics. The pure power-law behavior results in the asymptotic limit, when the number of informed agents in the network is large. The exponential behavior, on the other hand, is obtained in the asymptotic limit of the number of connection of each agent increasing indefinitely.

The locally transmitted information, which goes from one agent to its neighbours and involves a limited number of nodes, independent of the total number of agents in the network, was studied. Its spreading dynamics reveal that the number of informed agents increases according to a q-exponential function. From the statistical point of view, this result indicates that the Tsallis Statistics is the correct framework to investigate the scale-free network. The constraint on the number of contact of each agent implies in an increasing number of levels in

the network as the population increases. This is contrary to the small-word hypothesis [1, 2], where the number of levels in the network is fixed and the number of contacts increases.

The time interval for the transmission increases as the squared-root of the number of individuals in the network. The scaling properties, establishes a power-law condition to the probability of transmission of a piece of information by agent in the network. Our results can be easily tested in real or simulated data by checking the characteristics of the distributions.

Differential equations describing the information spread are derived. We discussed the different strategies one can take to increase the information spread efficiency. These strategies can be formulated by increasing the transmission efficiency or by the number of connected agents. We show that the most effective strategy is the second one.

The results obtained in the present work have an impact on the formulation of the best strategies for the information spread. In practice, it may have implications on Environment Sciences [41], since modifications in the local environment may evolve by locally transmitted effects to larger and distant areas [42–44]; Epidemiology, since viruses may evolve, by mutagenesis, to variants that optimize its transmission, a track that would prefer increasing the number of degrees of freedom by extending the period of virus transmission rather than increasing its transmission rate [45, 46]; Sociology [8], since communication among individuals in the society can be made more or less effective by controlling the mechanisms of spreading, what may have an impact in policies and strategies to, e.g., combat fake-news and other irrational behaviours in social media [47–50]. In Computer Sciences [15, 33], Biology [14], Physics [21], Economics [9], and Machine Learning [51], to name just a few.

The model of fractal network studied here can be modified in some aspects without changing the conclusions.

## Author Contributions

**Conceptualization:** Airton Deppman.

**Formal analysis:** Airton Deppman, Evandro Oliveira Andrade-II.

**Funding acquisition:** Airton Deppman.

**Investigation:** Airton Deppman.

**Methodology:** Airton Deppman.

**Project administration:** Airton Deppman.

**Validation:** Airton Deppman, Evandro Oliveira Andrade-II.

**Writing – original draft:** Airton Deppman.

**Writing – review & editing:** Airton Deppman, Evandro Oliveira Andrade-II.

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
