## [Decision Letter · Decision Letter 0]

26 Jul 2021

PONE-D-21-18197

Flux of information in scale-free networks

PLOS ONE

Dear Dr. Deppman,

Thank you for submitting your manuscript to PLOS ONE. After careful consideration, we feel that it has merit but does not fully meet PLOS ONE’s publication criteria as it currently stands. Therefore, we invite you to submit a revised version of the manuscript that addresses the points raised during the review process.

We recommend that it should be revised taking into account the changes requested by the reviewers. Since the requested changes include **Major Revision**, the revised manuscript will undergo the next round of review by the same reviewers.

We look forward to receiving your revised manuscript.

Kind regards,

Afnizanfaizal Abdullah, PhD. (Computer Science)

Academic Editor

PLOS ONE

Journal Requirements:

"AD would like to thank the support from the Conselho Nacional de Desenvolvimento Cient´ıfico e Tecnol´ogico (CNPq-Brazil) under grant 304244/2018-0, and from by FAPESP under Grant No. 2016/17612-7"

"AD, grant 304244/2018-0, Conselho Nacional de Desenvolvimento Científico e Tecnológico, http://www.cnpq.br, did not play any role in the design, analysis, prepration or submission of the work.

AD, grant 2016/17612-7, Fundação de Amparo à Pesquisa do Estado de São Paulo, https://fapesp.br, did not play any role in the design, analysis, prepration or submission of the work."

Reviewers' comments:

Reviewer's Responses to Questions

**Comments to the Author**

1. Is the manuscript technically sound, and do the data support the conclusions?

Reviewer #1: Yes

Reviewer #2: Partly

2. Has the statistical analysis been performed appropriately and rigorously? 

Reviewer #1: No

Reviewer #2: I Don't Know

3. Have the authors made all data underlying the findings in their manuscript fully available?

Reviewer #1: No

Reviewer #2: Yes

4. Is the manuscript presented in an intelligible fashion and written in standard English?

Reviewer #1: Yes

Reviewer #2: Yes

5. Review Comments to the Author

Reviewer #1: Title

- too general. please revise and be specific.

Abstract

- ok, acceptable. but not result/finding and conclusion. need to add these elements.

Introduction

- line 2 - you mention "large number of problems", what is the problems? it is good if you can give the example of problem.

- Authors only give the example of works (ref 5,15,13,16-18, 20, 21,22,23,24,11,25) use the scale symmetry in complex networks. It is good if authors discuss what the existing works do in brief and give the conclusion on the discussion.

- It is good if author can add the contribution/novelty of this work at the end of this section (before the organisation of this paper). This element missing in the paper.

- the problem in this domain (the scale symmetry in complex networks) is not clear. try to highlight what is the problem or research gap ini this section.

Section 1 - Properties of the scale-free network

- acceptable

Section 2 - Flux of information in a Fractal Network

- acceptable

Section 3 - Discussion of the results

- No results present in this section. Authors need to visualise the result (graph, table form etc).

Section 4 - Conclusion

- acceptable

Reviewer #2: The paper studies flux of information in fractal network, the mechanism to represent information spreading and statistical approach based on Tsallis statistics to investigate a scale-free networks. Differential equations also derived to describe information spreading. In general, the paper is interesting, well organized, and mathematically proof. However, the current form of the paper is not ready to reach on the global audience as it has major weakness, and the significant contributions is also not well highlighted. Please see following concerns:

1. The title should describe on overall research work. The current title is too general. It is not clear how the present work could potentially benefit from a wide range of natural and social phenomena.

2. The author should describe deeper for the application of scale-free networks based in every field so that the reader knows their study may be related to fractal networks or scale-free networks.

3. L13: It is stated that scale-free networks and fractal networks are the same. It’s pretty the same but has a slightly different. If necessary, give more examples of the actual case of scale-free and fractal networks to differential between two. The paper in [12] not mainly defined fractal networks as scale-free networks.

4. L29-L36: The field that related to symmetry scale in complex networks is identified. But this is too broad. The author should focus on the example for each area (same in Point 3)

5. L39: Other studies have presented the association of Tsallis Statistics and other statistical analysis in fractal networks. What is the main contribution to this work and different to others?

6. L56. Again, the author stated the equivalent of fractal and scale-free networks. The paper's title stated the scale-free, but throughout the work, the author uses the term fractal. A bit confusing.

7. L86. fnet shoud be italic

8. Put example one or two figures of the networks / nodes / connection to illustrate Definition 1.1 until Definition 1.7.

9. L127. Flux of information in a Fractal Network. Is same with the title?

10. Please re-check the equation and description. Some are not in standard form. For example, in L316, it is not usual to write ‘t0 is the instant when the spread of information starts’. What do you mean ‘instant’?

11. L332. Subsection is information diffusion, but the text is focus of optimization of the information spread to increase efficiency and communication.

12. On the other hand, this work may not be a very recent trend as there are many statistical approaches in another modern field, such as deep learning, neural networks, etc. The author should discuss some recent works in neural network or any related field related to this research in terms of the complexity of the networks.

6. PLOS authors have the option to publish the peer review history of their article (what does this mean?). If published, this will include your full peer review and any attached files.

Reviewer #1: No

Reviewer #2: No

---

## [Author Response · Author response to Decision Letter 0]

11 Aug 2021

Dear Editor,

Thank you for the opportunity to revise our submission according the reviewer’s suggestions. Below we describe how each of the points mentioned by the reviewer were considered and the corresponding revision in the new version of the manuscript. The boldface text corresponds to the modifications in the manuscript, were it is also displayed boldface. Only a modification involving several paragraphs do not appear in boldface, but is mentioned in the reply below.

Reviewer #1:

The paper studies flux of information in fractal network, the mechanism to represent information spreading and statistical approach based on Tsallis statistics to investigate a scale-free networks. Differential equations also derived to describe information spreading. In general, the paper is interesting, well organized, and mathematically proof. However, the current form of the paper is not ready to reach on the global audience as it has major weakness, and the significant contributions is also not well highlighted. Please see following concerns:

 1. The title should describe on overall research work. The current title is too general. It is not clear how the present work could potentially benefit from a wide range of natural and social phenomena.

The title was changed, and the new version brings “Emergency of Tsallis statistics in fractal networks” as the new title. With this title, we emphasize the relation with Tsallis Statistics. We keep the term scale-free, for the reasons discussed below.

 2. The author should describe deeper for the application of scale-free networks based in every field so that the reader knows their study may be related to fractal networks or scale-free networks. 

We modified the text, in the introductory part to a larger extent, but also in other parts of the manuscript, in order to make clearer the distinction between scale-free and fractal networks (see reply to the next point). We included more examples of fields of application and more citations to previous works. We also included, in the discussion, an example of application to the case of epidemic diseases.

 3. L13: It is stated that scale-free networks and fractal networks are the same. It’s pretty the same but has a slightly different. If necessary, give more examples of the actual case of scale-free and fractal networks to differential between two. The paper in [12] not mainly defined fractal networks as scale-free networks.

We agree with the referee that our statement was misleading. We modified the text in the introductory part in order to make a clear distinction between fractal and scale-free networks. The papers by Song et al. Are the basis we used for delineating the differences between the two types of network. We also included examples of fractal networks and new references.

These modifications are made in the following way: the third paragraph of the previous version of the manuscript was moved up to the second paragraph, and what was the second paragraph was extended and split in three new paragraphs. There we discuss in more details the scale-free network and the fractal network.

 4. L29-L36: The field that related to symmetry scale in complex networks is identified. But this is too broad. The author should focus on the example for each area (same in Point 3)

This part of the text was modified to a large extent, as mentioned already. The term scale symmetry was removed, and we mention self-similarity instead.

 5. L39: Other studies have presented the association of Tsallis Statistics and other statistical analysis in fractal networks. What is the main contribution to this work and different to others?

We included references to previous works associating Tsallis statistics and networks.

 6. L56. Again, the author stated the equivalent of fractal and scale-free networks. The paper's title stated the scale-free, but throughout the work, the author uses the term fractal. A bit confusing.

We modified the title, as already mentioned, and make clearer distinctions between scale-free networks and fractal networks.

 7. L86. fnet shoud be italic 

We wrote fnet in italic wherever it appears in the text.

 8. Put example one or two figures of the networks / nodes / connection to illustrate Definition 1.1 until Definition 1.7.

We included two figures in the manuscript. Fig.1 shows schematically the fractal structure of the agent based network. In Fig.2 we present a plot showing the dynamical evolution of the information diffusion process in the network.

In order to make a complete example of application of the information transmission model proposed here, we extended the calculations providing two new theorems, 2.6 and 2.7 in the new version.

 9. L127. Flux of information in a Fractal Network. Is same with the title?

Yes, but now the title was modified to mention fractal network instead of scale-free network.

 10. Please re-check the equation and description. Some are not in standard form. For example, in L316, it is not usual to write ‘t0 is the instant when the spread of information starts’. What do you mean ‘instant’?

To make the significance of t0 clearer, we modified the equations in the Corolary 2.5.2, and instead of using the multiplicative factor i_o we introduce a sum over all agents participating in the information spreading process. The meaning of the parameter $t_o$ (now indicated for each agent j as $t_{oj}$ is explained in more details. For the sake of clarity, we do not use the index j when we refer to a single agent in the subsequent equations. This is explained in a footnote.

 11. L332. Subsection is information diffusion, but the text is focus of optimization of the information spread to increase efficiency and communication.

We modified the title of the section to Strategies for optimization of the information diffusion

 12. On the other hand, this work may not be a very recent trend as there are many statistical approaches in another modern field, such as deep learning, neural networks, etc. The author should discuss some recent works in neural network or any related field related to this research in terms of the complexity of the networks.

We included one example on epidemic dynamics, and we mention connections with thermodynamical systems and with high energy physics, which is closer to the line of research we are following.

We cite references on fractal networks applications to WWW and AI.

We also corrected typos and errors fond in Theorem 2.10, which is now extended to include more steps of the proof, for the sake of clarity.

---

## [Decision Letter · Decision Letter 1]

13 Sep 2021

Emergency of Tsallis statistics in fractal networks

PONE-D-21-18197R1

Dear Dr. Deppman,

We’re pleased to inform you that your manuscript has been judged scientifically suitable for publication and will be formally accepted for publication once it meets all outstanding technical requirements.

Kind regards,

Afnizanfaizal Abdullah, PhD. (Computer Science)

Academic Editor

PLOS ONE

Additional Editor Comments (optional):

Reviewers' comments:

Reviewer's Responses to Questions

**Comments to the Author**

1. If the authors have adequately addressed your comments raised in a previous round of review and you feel that this manuscript is now acceptable for publication, you may indicate that here to bypass the “Comments to the Author” section, enter your conflict of interest statement in the “Confidential to Editor” section, and submit your "Accept" recommendation.

Reviewer #2: All comments have been addressed

2. Is the manuscript technically sound, and do the data support the conclusions?

Reviewer #2: Partly

3. Has the statistical analysis been performed appropriately and rigorously? 

Reviewer #2: I Don't Know

4. Have the authors made all data underlying the findings in their manuscript fully available?

Reviewer #2: No

5. Is the manuscript presented in an intelligible fashion and written in standard English?

Reviewer #2: Yes

6. Review Comments to the Author

Reviewer #2: The author has made necessary amendments based on the previous issues. The manuscript still need to checked the english and some error need to be fixed (e.g citation error at L79, L83).

7. PLOS authors have the option to publish the peer review history of their article (what does this mean?). If published, this will include your full peer review and any attached files.

Reviewer #2: No

---

## [Editor Report · Acceptance letter]

20 Sep 2021

PONE-D-21-18197R1 

Emergency of Tsallis statistics in fractal networks 

Dear Dr. Deppman:

I'm pleased to inform you that your manuscript has been deemed suitable for publication in PLOS ONE. Congratulations! Your manuscript is now with our production department. 

Kind regards, 

on behalf of

Dr. Afnizanfaizal Abdullah 

Academic Editor

PLOS ONE